# Assessing reactive violence using Immersive Virtual Reality

**Sylvia Terbeck**[1]*, **Chloe Case**[1], **Joshua Turner**[2], **Victoria Spencer**[2], **Alison Bacon**[2], **Charles Howard**[2], **Ian S. Howard**[3]

**1** School of Psychology, Liverpool John Moores University, Liverpool, United Kingdom, **2** School of Psychology, University of Plymouth, Drake Circus, Plymouth, United Kingdom, **3** SECAM, University of Plymouth, Drake Circus, Plymouth, United Kingdom

* s.terbeck@ljmu.ac.uk

## Abstract

Assessing levels of aggression–specifically reactive violence–has been a challenge in the past, since individuals might be reluctant to self-report aggressive tendencies. Furthermore, experimental studies often lack ecological validity. Immersive Virtual Reality (IVR) offers a reliable, ethically safe environment, and is the most realistic virtual simulation method currently available. It allows researchers to test participants' aggressive responses to realistic provocations from virtual humans. In the current study, 116 participants completed our IVR aggression task, in which they encountered avatars who would either approach them in a friendly or provocative fashion. Participants had the option either to shake hands or hit the virtual human, in congruent and incongruent trials. In congruent trials, the response required of the participant matched the approach with the avatar (e.g., hitting the avatar after provocation). In incongruent trials there was a mismatch between the avatars approach and the participants required response. Congruent trials were designed to measure the immediate reaction towards the virtual human, and incongruent trials to assess response inhibition. Additionally, participants also completed traditional questionnaire-based measures of aggression, as well as reporting their past violent behaviour. We found that the immediate aggressive responses in the IVR task correlated with the established questionnaire measures (convergent validity), and we found that the IVR task was a stronger predictor of past violent behaviour than traditional measures (discriminant validity). This suggests that IVR might be an effective way to assess aggressive behaviour in a more indirect, but realistic manner, than current questionnaire assessment.

## Introduction

The Crime Survey for England and Wales recorded 1.2 million incidents of violence in 2020 [1]. There is thus an urgent need to further investigate and understand the causes of such violence, in order to come up with a means to reduce it. Aggression can be defined as any intentional behaviour to harm another person who does not want to be harmed [2].

Displays of aggression can be divided into impulsive, which is affective and reactive, and instrumental, which is proactive [2]. It has been suggested the impulsive subtype is related to

**Data Availability Statement:** The data is available at this LINK: https://osf.io/cmhyx/?view_only=6050204907e84cd59c6d0281af179be6.

**Funding:** S.T. Grant number SG162648 British Academy https://www.thebritishacademy.ac.uk/?gclid=Cj0KCQjwvaeJBhCvARIsABgTDM6raFCe

JzNRNaDU253xCyXH0wQ4irkfBqr8GWj
EGGv54hEPfTWUTSkaAkKmEALw_wcB The
funder played no role in the research.

**Competing interests:** The authors have declared
no competing interest exist.

poor emotion regulation and poor impulse control [3], which is especially present in situations in which individuals are provoked socially [4]. Work by Billen et al. [5] determined that crime and recidivism (the tendency of a convicted criminal to reoffend) were strongly associated with levels of self-control, which involved factors of coping skills as well as impulsivity. Indeed, Gottfredson and Hirschi [6] proposed self-regulation/self-control to be the strongest predictor of crime and aggressive acts. Furthermore, triggers for aggressive behaviours may involve discrete social dynamics [7] and in order to produce a none-aggressive response to another person, some individuals might activate inhibition and self-control to consciously inhibit their aggressive behavioural impulses.

Previously, observation and self-report have been the most common way to assess levels of aggression in individuals. Specifically, researchers have most often implemented measures of the variables that are thought to underlie and precipitate aggression such as irritability, hostility, impulsivity, and anger [8–10]. For instance, to assess facets of aggression, Buss and Perry [9] developed the Aggression Questionnaire (AQ) as a refined and expanded version of their earlier Buss-Durkee Hostility Inventory (1957). Probably the most widely used assessment, The State Trait Anger Expression Inventory-2 (STAXI-2) [11] uses manifested anger and an individual's self-reported expression of it as a measurement of aggression. Using the STAXI-2 as a predictor for aggression is common, due its potential ability to predict some form of aggressive behaviours, such as institutional aggression [12]. However, studies investigating the relationship between the STAXI score and observed aggressive behaviour, have shown mixed results. For instance, Cornell, Peterson, and Richards [13] found that it failed to predict previous aggression in juvenile offenders, but that it was successful in predicting future aggression at the 3-month follow-up. Additionally, it was found that the STAXI subscale 'state-anger' was not correlated with aggressive behaviour [14]. In forensic settings, it has been found that the STAXI highly correlates with impression management (IM; i.e., giving answers which are perceived to be socially acceptable/desirable), whereby patients engaging in IM self-report significantly less levels of outward, and inward anger and higher levels of anger control [15]. Indeed, implementation of self-reported measures of aggression can lead to biased responses from participants due to the influence of social desirability [16]. Bech and Mak [17] evidenced an inverse relationship between measures of social desirability and hostility, whereby participants motivated by social approval reported less hostility than those who placed less value on social acceptance. Furthermore, Suris et al. [18] stated that utilizing self-report indicators of aggression often proves difficult as participants frequently share elements of higher order constructs and therefore are interrelated to the point of sharing common variance. Samples used in previous studies often implement contrastingly different assessments of the constructs of violence and aggression and therefore this might contribute to the low correlations within criterion measures. Recently, Berlin et al. [19] determined that self-reports of physical aggression where highly correlated with clinician's assessment of aggression. However, both were not related to violent offence records.

Noting the potential limitations of self-report and introspective measures, researchers have developed studies of aggressive behavioural acts, involving tasks to induce frustration within the laboratory. Instruments include continuous performance tasks (CPT) such as the Integrated Visual and Auditory CPT [20] and interactive provocation tasks such as the Point Subtraction Aggression Paradigm Task [21] all suggested to potentially induce frustration. The Hot Sauce Paradigm [22] used measurements of hot sauce knowingly applied to another person's food by a participant (where the consumer disliked the hot sauce) as a behavioural index of aggression in participants. Lieberman et al. [22] claimed this method overcomes limitations of other behavioural paradigms because participants perceive the potential for real harm to come to the target person. However, the extent to which hot sauce administration can be

regarded as hostile physical aggression, and whether such laboratory studies hold ecological validity that is generalizable to a real-world setting, is debatable [23].

More recently, Verhoef et al., [24] developed the first pilot study, assessing children's levels of aggression using virtual reality. In this study, 32 children aged 8–13 years were immersed in a virtual classroom in which they encountered several scenarios. For instance, another virtual child might push over a large toy tower as an act of provocation. The researchers found that the Immersive Virtual Reality (IVR) experience was greatly enjoyed by the children and that its behavioural observation results correlated strongly with the traditional paper pencil assessment of aggression. They also found that the IVR measure showed higher discriminant validity by being able to better determine individual differences in the level of aggression in children.

Virtual Reality was previously more commonly used in studies as an intervention tool to reduce aggression, with the hope that it could potentially be used as an intervention in clinical practice in the future. In a recent review, Dellazizzo et al., [25] examined previous studies, which have developed virtual reality interventions to reduce violence in incarcerated youth, in particular focussing on youth offenders who suffer from schizophrenia. They found few studies who have used IVR as an intervention tool for aggression but reported that preliminary studies using IVR showed some promising results, by reducing anger, as well as improving conflict resolution skills in teenagers. For example, Smeijers and Koole [4] developed a study protocol for an upcoming study, using an IVR game as an add-on tool to manage and reduce aggression, by using avoidance movement training to angry faces in forensic psychiatric outpatients. In IVR, virtual humans meet the patients in a virtual shopping street, approaching them in either a friendly or unfriendly manner. In the task the patients are then instructed to lean forward to agreeable avatars and lean backwards to disagreeable avatars. The authors speculate that this training now delivered in IVR will more significantly help to reduce anger and aggressive behaviour in the patients.

Klein Tuente et al., [26] applied virtual reality aggression prevention therapy to a sample of 28 offenders inpatients who displayed aggressive behaviour. The training involved 16 one-hour sessions in IVR, in which participants encountered adult virtual humans provoking them. The authors found that the participants' self-reported or observed aggression was not reduced by the training. They did though find that the participants' hostility and self-control significantly improved, although the authors reported that these changes were not sustained at the 3-month follow up.

Our study investigated self-control and response inhibition in association with aggressive behaviour, using a novel IVR task to assess levels of reactive aggression. The use of IVR in recreating simulated social scenarios has been demonstrated to elicit realistic responses by human participants; equivalent to real world responses [27]. Rovira, Swapp, Spanlang, and Slater [28] proposed IVR to be an effective measure in the study of violent situations, with high ecological validity. Furthermore, Rovira et al. [28] suggested that scenarios in IVR can elicit realistic responses via engagement of participants if the issue of plausibility is addressed correctly. Plausibility encompasses not only the credibility of the scenarios presented to participants, but also that their actions within the virtual world can have appropriate responses and are recognised within the illusionary virtual world. These issues of plausibility are addressed through sensorimotor contingency and domain design, creating a pragmatic method of measuring acts of aggression. The virtual human avatars used in our IVR task needed to act in a realistic manner (i.e., where possible exhibit random body movements, synchronizes animated moving lip movements when speaking, and make mild gestures that as closely as possible mimic the natural physical responses observed in real humans). Finally, similar to Smeijers and Koole [4] we also included elements of the 'approach avoidance modification paradigm' (i.e., forward

movement as indicator of approach/attack and backwards movements as an avoidance/inhibition measure).

The overall aim of our study was to test our new IVR task as an assessment to measure levels of reactive aggression. We propose the hypothesis, that this new task would show convergent validity to traditional measures and that it would be a strong predictor of past violence. Specifically, we investigated if the response times to provocation by virtual humans (i.e., time to exhibit aggressive responses following the provocation) were related to the level of self-reported aggression. This suggests that the faster someone would exhibit the aggressive response after provocation, the more accessible this response option might be to that person, suggesting potential more impulsive aggressive tendencies. Furthermore, we examined if IVR response times were a better predictor of past violence than self-reported measures of aggression.

## Method

### Participants

Participants were recruited through the University's online participation system, which provided a sample of participants who were mainly students, but included other non-student individuals who had voluntarily signed up to the university's participants system. Out of the 116 participants ($M_{age}$ = 25.91, $SD$ = 12.43; 30 male) who took part in this study, 6% of the participants had attained education to a GCSE level, 72.4% to an A-level, 19.8% had a university undergraduate degree, and 1.7% had postgraduate education. Participants were reimbursed with either financially reward or with course credit. The study received ethical approval from Plymouth University ethics committee and we obtained written consent from every participant.

### Materials

**The State-Trait Anger Expression Inventory-2 (STAXI-2) [11].** The State-Trait Anger Expression Inventory (STAXI-2) included 57 items on six scales. The six scales were: state-anger, trait-anger, anger expression-out, anger expression-in, anger control- out and anger control-in. A sample item was "*I control my urge to express my angry feelings*." For state anger, ratings were given on a 4-point scale (ranging from 1 = *not at all* to 4 = *very much so*). Trait anger measured the predisposition an individual possesses to become angry; the ratings were also given on a four-point scale (ranging from 1 = *almost never* to 4 = *almost always*). Anger expression was measured by the anger expression index, which evaluated the person's propensity to express anger either outwardly to others or inwardly to themselves. Anger control indicated the level to which an individual prevents angry feelings by reducing them being expressed towards the other person or by reducing them internally. It was noted that the internal consistency of the subscales varied from α = .82 to α = .75 [11] (In this α = .72; α = .83 for state anger and α = .57 for trait anger; α = .54 for anger expression, α = .74 for anger control).

**Reactive Proactive Aggression Questionnaire (RPAQ) [29].** This is a 23 item self-report questionnaire to assess reactive (following provocation) and proactive (instrumental) past aggression. Items are rated from 0 = *never* to 2 = *often*. A sample item for reactive anger includes "*Became angry when others threatened you*". Cronbach's alphas for the reactive and proactive scales were reported to be of 0.84 and 0.86, respectively.

**The immersive virtual reality task.** An Oculus Rift headset and a pair of Sennheiser headphones were used for the IVR task. The IVR program ran on a desktop PC and the participant was seated at a computer desk. There was a 22" LCD computer monitor located immediately in front of the participant so the experimenter could also monitor task progress. A

(a)

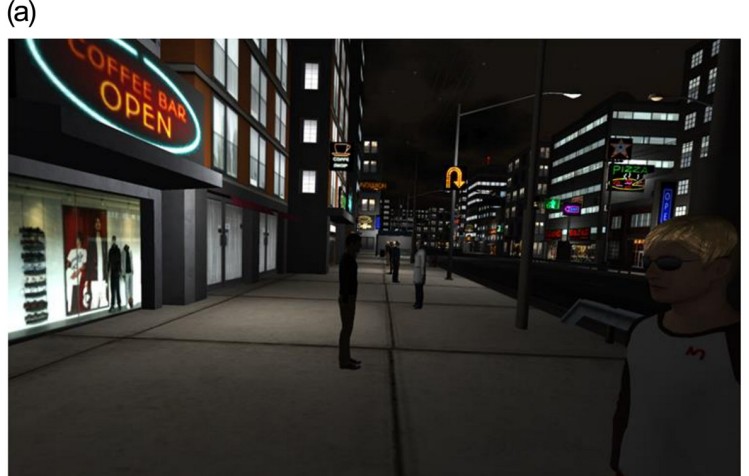

(b)

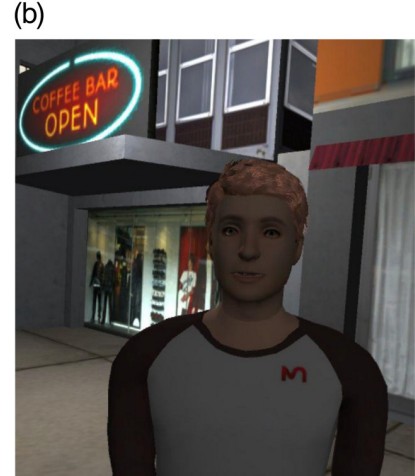

(c)

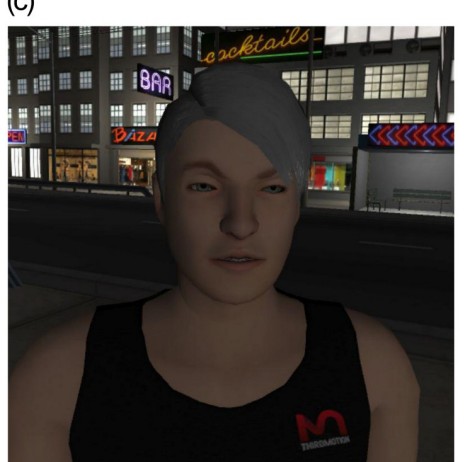

(d)

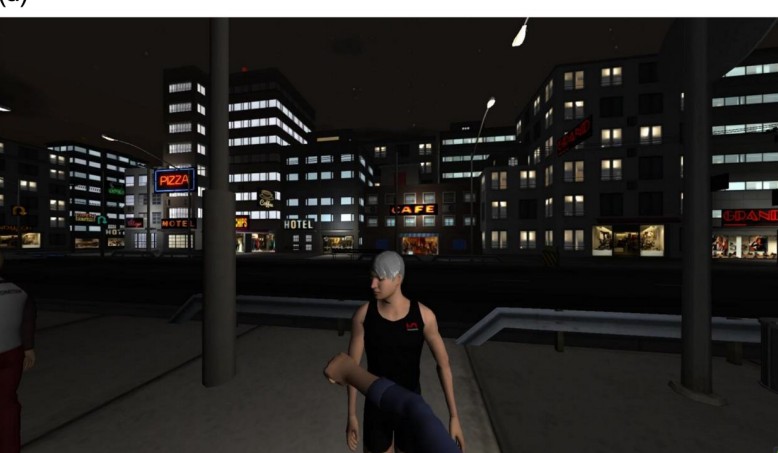

**Fig 1. Illustrations of the IVR task.** Panel A shows the street that the participant's avatar walks down. Panel B & C Close up of the virtual agents that require a response from the participant. Panel D Participant generating an aggressive action from their avatar and hitting a victual agent.

joystick was used by participants to give the relevant responses and speed of response and the type of the responses were automatically recorded by the program. In the task individual avatars greeted the participant either pro-socially such as "*Hello, how are you*?" or non-pro-socially e.g. "*What are you looking at*?" on a hypothetical nightlife city street (See Fig 1). The task consisted of two different sets of 15 trials, the first set requiring congruent social reactions and the second set requiring incongruent social reactions. During the experiment, tasks were selected from the two lists at random. In the congruent trials condition, participants were required to push the joystick forward, to punch unfriendly avatars following the interaction, or pull the joystick backwards, to initiate a handshake with friendly avatars. In the incongruent trials condition, participants were required to react in the opposite way, thus an avatar interacting in a friendly manner would require a proceeding 'punch' response by participants (joystick moved forward) and an interaction with an unfriendly avatar would require a proceeding reaction from the participant in the form of a handshake (joystick moved backwards) (See Table 1).

**Table 1. Blocks in the IVR task.** The table rows show the 4 experimental conditions. The condition column indicates the 4 possible combinations of anti-social and pro-social behaviour in congruent and incongruent scenarios. Note that we termed both responses to the incongruent trials as response inhibition due to the incongruent nature of the responses. The Avatar behaviour column shows anti-social avatar behaviour is aggressive, whereas pro-social behaviour is friendly. The Participant's Instruction to Act column indicates how the participant is asked to behave for each experimental condition. The construct to assess column indicates the relevance of the response time in each condition.

| Condition | Avatar Behaviour | Participant's Instruction to Act | Construct assessed |
|---|---|---|---|
| **Anti-Social 1 (Congruent)** | Aggressive | Aggressive | Response time: Aggressive response following provocation |
| **Pro-Social 1 (Congruent)** | Friendly | Friendly | Response time: Friendly response following friendly approach |
| **Anti-Social 2 (Incongruent)** | Aggressive | Friendly | Response time: Response inhibition following provocation |
| **Pro-Social 2 (Incongruent)** | Friendly | Aggressive | Response time: Response inhibition following friendly approach |

The program recorded the response action as well as the response time from when the avatar started speaking until the participant started to move the joystick. The avatars voices were all male pre-recorded and lip-synced to the virtual human to create a realistic impression. Apart than the lip sync, there were no expressions of anger or approach in the avatars face or posture. The participant was embodied into a virtual body with a virtual arm, which enacted the reaction made towards the avatar.

To enable the statistical analysis of the IVR data, a file was created with the response times recorded over the trials. For all trials in each of the four conditions in Table 1, the average response time was calculated. For instance, for the 15 trials in condition 1, the average response time for that condition was then created. All condition's average response times were then used for the correlational analysis. We hypothesise that shorter reaction times in the Anti-Social 1 condition (e.g., faster responding to provocation) indicated greater level of aggression.

## Procedure

After informed consent, participants first completed the IVR task in the IVR laboratory followed by the questionnaires. This task order was adopted to avoid participants focussing on the assessment of aggression during the IVR task. Participants were informed that the IVR might be distressing and that they could end the procedure at any time if they felt uncomfortable. No participant reported that they felt uncomfortable and no participant left.

## Results

### General response tendencies

Firstly, we investigated the general response patterns within each block, calculating differences in response time overall between the conditions (See Fig 2).

Fig 2 shows the mean response times in the different conditions, suggesting that participants were generally faster to respond to anti-social avatars. We conducted a repeated measures ANOVA with condition as the repeated measures factors and found it to be significant $F_{(2.74)} = 92.73$, $p < .01$. Greenhouse Geisser corrected post-hoc tests revealed significant differences between all conditions (all $ps < .01$).

Overall, 63% of participants made no errors at all in the congruent condition and 53.4% in the incongruent condition. Only a relatively small proportion of 13.8% of participants made more than 1 error in the incongruent condition and 19.8% in the congruent condition. We found that errors were not normally distributed ($D(116) = .30$, $p < .01$). A Wilcock test revealed that participants made significantly more errors in the congruent compared to the incongruent condition ($Z = -2.86$, $p = .004$).

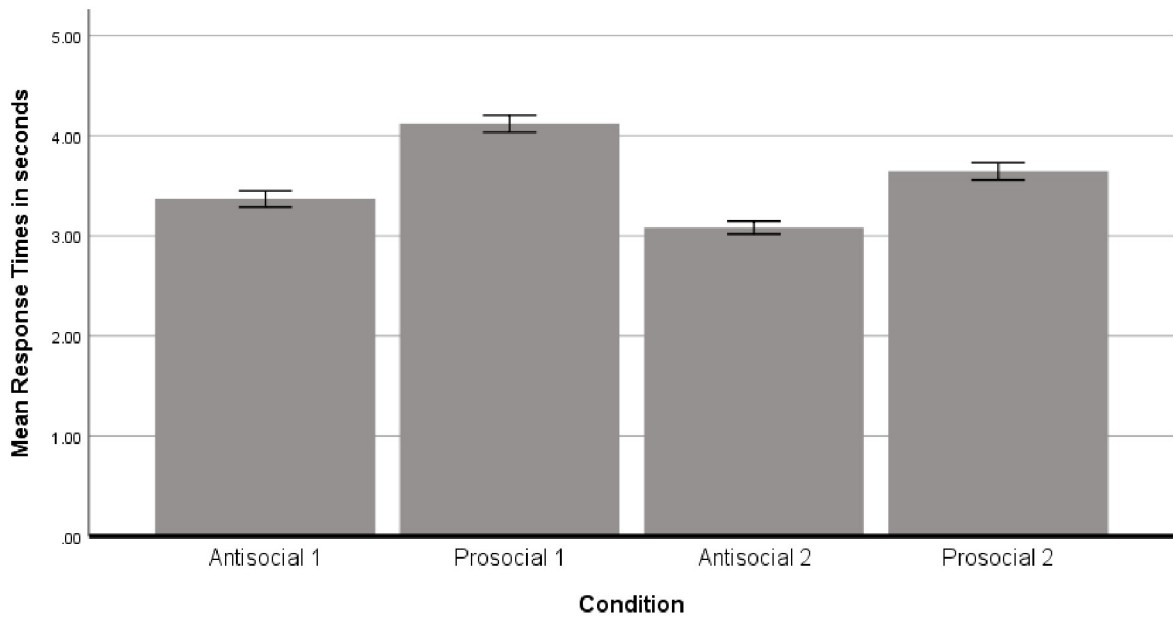

**Fig 2. Response time differences during the different conditions in the IVR task.** Bar height indicates the mean response time and the error bars indicate +/− the standard error.

## Correlations with individual differences measures

Table 2 illustrates the Pearson correlations between the IVR task and the traditional measures. Aggression in the anti-social 1 condition showed a significant negative correlation with the total anger expression index (AX) of the STAXI-2, as well as with the reactive past aggression self-report. Furthermore, the self-report STAXI-2 scales also correlated with past reactive aggression.

We used Spearman's correlation to determine none-parametric correlations with aggression self-report measures and errors made in the IVR task, however, we did not find any significant relationships (all $ps > .05$).

**Table 2. Pearson r correlations between IVR task and aggression measures.**

| | Pro-Social 1 | Pro-Social 2 | Anti-Social 2 | Stage Anger | Trait Anger | AX | RPAQ | Pro-active | Reactive |
|---|---|---|---|---|---|---|---|---|---|
| **Anti-Social 1** | .68** | .63** | .72** | -.08 | -.15 | 25** | -.20* | -.10 | -.22* |
| **Pro-Social 1** | | .55** | .67** | .04 | -.06 | -.13 | -.80 | .01 | -.15 |
| **Pro-Social 2** | | | .79** | -.01 | -.01 | -.09 | -.14 | -.12 | -.18 |
| **Anti-Social 2** | | | | -.02 | -.07 | -.17 | -.13 | -.06 | -.17 |
| **State Anger** | | | | | .31** | .15 | .34** | .16 | .09 |
| **Trait Anger** | | | | | | .71** | .62** | .06 | .26** |
| **AX** | | | | | | | .54** | .01 | .14 |
| **RPAQ** | | | | | | | | .35** | .47** |
| **Proactive** | | | | | | | | | .52** |

** is significant at .01 level;

* is significant at .05 level

**Table 3. Factors in the regression model.**

| Concept | B | T |
|---|---|---|
| Constant | 6.39 | 2.89* |
| Trait anger | .20 | 2.6** |
| IVR = Anti-Social 1 (congruent 1 condition) | -.94 | -2.1* |

Finally, using backward regression, the final remaining model, including Trait Anger and IVR Anti-Social 1 (congruent condition) could significantly predict reactive past aggression. Model R Square = .104; *p* = .002 (See Table 3).

## Discussion

The aim of this study was to investigate a novel, interactive IVR task to assess levels of reactive aggression after provocation by virtual humans. In particular, we compared responses in the IVR task to traditional self-report-based measures of aggression and its relationship to aggressive past behaviour. We found that the response time towards virtually hitting an avatar after provocation correlated with the overall anger expression index of the STAXI-2, as well as reactive aggression from the proactive-reactive past aggression questionnaire. Specifically, as predicted by our hypothesis, we found that only the response times to act aggressive following provocation (i.e., not the response times in all conditions) were related to self-reported levels of aggression. Thus, higher levels of self-reported aggression were correlated to lower reaction times on the IVR paradigm. Furthermore, we found that this IVR task component was a better predictor of past violent reactive self-reported aggression than the self-report measures.

These findings support previous IVR measures, who also started to find associations between IVR aggressive responses and traditional self-report measures [24]. Verhoef et al., [24] developed the first IVR pilot assessment for children and also found good convergent and discriminant validity for the task. Indeed, establishing IVR as a valid measure of aggression has important implications for future uses of this advancing technology. Rovira et al. [28] concluded that IVR may also be useful for rehabilitation of victims of aggression who might become disturbed by their behavioural response in a real-world situation. With further research, this conclusion could be expanded to include aggressors themselves who might benefit from the realisation of not only their actual behaviour, but also the consequences to the recipients and victims of their aggressive behaviour. Identifying risk factors and protective factors of aggression could allow for appropriate strategies and frameworks of intervention to be implemented more effectively, potentially yielding more successful outcomes for individuals, rather than interventions occurring reactively to violent or other types of aggressive behaviour.

IVR delivers an opportunity to measure immediate emotional responses [28], and in our task we suggest that the response time to display the aggressive response after provocation can be used an indicator of an individual's accessibility of the aggressive emotional impulses in the 'heat of the moment'.

Participants however, made also errors in the IVR task, which constituted making the opposite response to the one required (e.g., they were hitting the virtual human when a handshake was required). Errors in response will clearly add noise to the results, but should not affect the underlying trends, especially when their incidence is low, as in our study. To reduce the effect of such errors, future studies could investigate disabling the option to perform the opposite response on the IVR response device during the task, thereby eliminating the opportunity to make an inappropriate response.

Overall, a few participants (13.8% in incongruent and 19.8% in the congruent) made more than one error, and the number of errors were not correlated to the self-report measures. We found that people made more errors in the congruent condition compared to the incongruent one. This might indicate that the congruent condition was processed by participants in a more emotional manner (i.e., with an intuitive and impulsive response after the virtual human's approach). In contrast, they might have engaged a more cognitive approach in the incongruent condition, in which they were concentrating more about providing the correct response rather than acting on impulse.

## Limitations

Our current study involved mainly students, of similar age and educational background, with the majority of them being female. Physical violence has however been reported to be greater in men [2], thus future studies would need to include more male participants to further validate the IVR task.

We used our current sample of participants as the first attempt to validate our new IVR measure. Future studies are clearly needed to assess and replicate the findings using different samples of participants. Thus future research could also involve the IVR task being performed on a sample of clinical interest, such as violent offenders. Previous research has proposed and demonstrated that behavioural measures perform better in populations where the targeted variable occurs at a higher rate of incidence and magnitude [30]. Direct comparisons of sample characteristic would allow for further analysis of the relationship and constructs of aggression and executive functioning, where clinically relevant features such as psychopathy or violent tendencies may moderate the relationship between self-control and aggression. Such comparisons give potential for an accepted taxonomy of anger and aggression and resolve the issues raised by Lee and DiGiuseppe [31] by better informing targets of treatment and rehabilitation.

A further limitation of the current study is the nature of self-report measure for past reactive aggression, as social desirability potentially confounds the self-reported measurements of aggression and anger. Future studies might implement observational methods in forensic settings as a variable for discriminant validity. Furthermore, in our IVR task, the responses were made using a joystick. In order to increase ecological validity of the task even more, other devices (i.e., also those simulating force) could be used to make the hitting/shaking hands movement not only visually, but also physically more realistic. Indeed, in a recent study using a morality IVR task, we implemented a robotic maniulandum to more realistically produce measures of force and physical touch into the task [32]. Additionally, order effects of the IVR task and presenting the questionnaire could be investigated in future studies.

We suggest our current study provides a framework and data for future research using the IVR Aggression Task. The future possibilities and benefits of IVR as an effective tool in psychological research are promising. Additionally, IVR is usually regarded as an engaging and 'fun' tool, thus making it potentially more attractive for participants to engage with the technology. In addition, IVR technology affords ecological validity without compromising experimental control, thereby increasing experimental impact, and reducing replication difficulties. Furthermore, since the tasks investigates aggressive responses to virtual humans, it does not involve aggressive responses to real humans, making it more ethically safe and acceptable. The advancement in IVR Technology brings great potential for developments in psychological experimentation, theory, and interventions. Thus, since it has been suggested that reducing reactive aggression is of top priority in forensic settings [26], effectively measuring it might be a first step to help to develop evidence-based interventions to reduce it. We thus believe this new IVR task a first step in its application *as an assessment tool* for levels of aggression in

forensic populations and to also to indirectly measure potential improvements after aggression treatment. This could be achieved by examining differences in our IVR task scores before, during and after an intervention. Future studies might also investigate the use of IVR at additional time intervals before during, after treatment. Additional measurements could also be added, combining the IVR assessment with direct physiological measures, such as heart rate and skin conductance.

Finally, we note that a presentation of some aspects of this work is available in an online YouTube video here: https://www.youtube.com/watch?v=xqPJKtFI42U

This was recorded at a TEDx talk in 2018 with the title: "The light and dark of emotional intelligence".

## Author Contributions

**Conceptualization:** Sylvia Terbeck, Alison Bacon, Charles Howard, Ian S. Howard.

**Data curation:** Joshua Turner, Ian S. Howard.

**Formal analysis:** Joshua Turner, Victoria Spencer.

**Funding acquisition:** Sylvia Terbeck.

**Investigation:** Alison Bacon.

**Methodology:** Joshua Turner, Charles Howard, Ian S. Howard.

**Project administration:** Chloe Case.

**Resources:** Chloe Case.

**Software:** Joshua Turner, Ian S. Howard.

**Supervision:** Sylvia Terbeck, Ian S. Howard.

**Validation:** Chloe Case.

**Visualization:** Chloe Case, Charles Howard.

**Writing – original draft:** Victoria Spencer.

**Writing – review & editing:** Chloe Case, Alison Bacon.

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
