## [Decision Letter · Decision Letter 0]

3 Dec 2021

PONE-D-21-27929Assessing reactive physical violence using Immersive Virtual RealityPLOS ONE

Dear Dr. Terbeck,

Thank you for submitting your manuscript to PLOS ONE. After careful consideration, we feel that it has merit but does not fully meet PLOS ONE’s publication criteria as it currently stands. Therefore, we invite you to submit a revised version of the manuscript that addresses the points raised during the review process.

Three Reviewers evaluated the manuscript. They were in general supportive of the manuscript, but major revisions are necessary. I suggest Authors to provide adequate responses and modifications, especially in terms of clarifications to background and methodology, and interpretation of results. ==============================

We look forward to receiving your revised manuscript.

Kind regards,

Stefano Triberti, Ph.D.

Academic Editor

PLOS ONE

Journal Requirements:

3. We note that you have stated that you will provide repository information for your data at acceptance. Should your manuscript be accepted for publication, we will hold it until you provide the relevant accession numbers or DOIs necessary to access your data. If you wish to make changes to your Data Availability statement, please describe these changes in your cover letter and we will update your Data Availability statement to reflect the information you provide

Reviewers' comments:

Reviewer's Responses to Questions

**Comments to the Author**

1. Is the manuscript technically sound, and do the data support the conclusions?

Reviewer #1: No

Reviewer #2: Partly

Reviewer #3: Yes

2. Has the statistical analysis been performed appropriately and rigorously? 

Reviewer #1: I Don't Know

Reviewer #2: Yes

Reviewer #3: Yes

3. Have the authors made all data underlying the findings in their manuscript fully available?

Reviewer #1: Yes

Reviewer #2: Yes

Reviewer #3: Yes

4. Is the manuscript presented in an intelligible fashion and written in standard English?

Reviewer #1: Yes

Reviewer #2: Yes

Reviewer #3: Yes

5. Review Comments to the Author

Reviewer #1: This manuscript described a novel IVR task, designed to assess aggression, and preliminary findings on its validity. Although I do concur with the authors that IVR provides completely new possibilities to assess (and treat) aggression, the manuscript requires substantial revisions before publication can be considered. I would love to see a revised version. Please see detailed suggestions and comments below.

Introduction

In general, the introduction needs to be revised and the authors must make sure to increase clarity in conceptualizations in relation to aggression. As it stands now, the authors shortly mention key terms that are used very broad but would require more context. Please make sure to be specific and not overgeneralize, since this may lead to crucial misinterpretations. There are several, recent references which could be used to improve the introduction. For instance, see Billen et al., 2019; Berlin et al., 2021; Martens et al., 2019.

p. 3, paragraph 1:

* Please provide the context for this statement, since it otherwise is too broad: "The prevalence of aggressive and violent acts is significant and requires attention and investigation."

* What do the authors mean by "Often dichotomous, violent acts arising from

aggression may be predatory, impulsive, reactive, or defensive and such behaviours may arise

from situational or environmental factors.". Please clarify what is meant by "dichotomous" in this respect.

p. 4, paragraph 1:

* This sentence comes without context, please remove or provide context: "Our novel IVR aggression task allows us to test levels of response inhibition in complex virtual social environments."

p. 4, paragraph 2:

* The AQ does not specifically target only hostility, the current phrasing can be misleading and I this recommend revision.

p. 5, paragraph 2:

* CPT tests are not conceptualized as aggression measures, which is how it may be read as is written currently. It may, however, induce frustration. Please remove or revise.

p.6, paragraph 2:

* VR is not a common intervention tool in aggression management, but is increasingly being tested as such in different forensic settings.

* As is written now, the reference to Klein Tuente (please revise to Klein Tuente) et al seems to refer to a similar intervention as Smeijers and Koole (VR-GAIME; which is only used as an add-on in ART trainings). These are two completely different methods. I recommend to either provide details for both, or just present shortly and as two independent methods.

p.7, paragraph 2: I strongly recommend the authors to not suggest that they will test causality, since this would require a completely different research design than what is provided in the manuscript at hand.

p. 7-8: I recommend the authors to, clearly, write the scientific questions that are being addressed within the current study, so that this part can be compared to how the results are presented.

Methods

p. 8: please provide more information on the participants.

p. 8: please provide internal consistency values for the STAXI-2 scales in the current study (or have I misinterpreted the values as coming from the reference?).

p. 8: I would suggest the authors to use "instrumental" instead of "forward" when referring to proactive aggression.

Please provide a part describing the statistical considerations and analyses.

Results

p. 15: what measure in the IVR task was used for the correlational analyses? As it stands now, the reader has no possibility to interpret the values provided by the table. This interpretation could be facilitated by a part in statistical analyses in the methods section. This information is provided in the discussion, but must be clear already in the methods and results sections.

p. 16: the headline for Table 2 should refer to Table 3, and be followed by Table 3, I guess?

Discussion

My main concern on this part is that the authors use response time on the IVR task as a proxy for aggression in IVR. Unfortunately, I do not see any solid evidence-base for this assumption anywhere in the manuscript. I strongly recommend the authors to revise the aims of the study (p. 7-8) to better correspond to what was actually tested in the study, and rewrite the discussion with this in mind.

Minor comments:

* please proof-read once more, some words are missing in some sentences, misspellings occur (e.g. anti-social, pro-social, some references), and abbreviations should fist be introduce and then used consequently throughout the manuscript.

Reviewer #2: First of all, I want to congratulate the authors on their study. This manuscript concerns an important issue for society, using innovative and promising techniques. The possibility of finding more ecologically valid instruments is very interesting, for both empirical research and clinical practice. That is why I enjoyed reading the manuscript.

I have some major and minor issues with the manuscript in its present form, however, which in my opinion need to be addressed before it is suitable for publication. These are listed below.

1. In the abstract, it is not clear enough what the function is of the congruent and incongruent trials. Could this be explained in a few words?

2. In the Introduction, concerning the following sentence: “Often dichotomous, violent acts arising from aggression may be predatory, impulsive, reactive, or defensive and such behaviours may arise from situational or environmental factors.” For readers who are not familiar with aggression concepts it may not be clear what ‘dichotomous’ refers to, especially since multiple terms from different models/authors are subsequently mentioned randomly. Please explain the dichotomous character. Furthermore, be aware that terms of different authors are mixed together in this sentence. Maybe it is preferable to stick to one model (for example: proactive versus reactive aggression according tot Dodge, or instrumental versus impulsive aggression). And, finally, add a reference here.

3. In the introduction, the general textual structure is difficult to follow and it is too lengthy. For example, in the first part of the introduction, the authors shift from prevalence to dichotomous aggression types, to models (with in between again dichotomous aggression types), to (yet again) similar aggression subtypes, followed by – out of the blue- an IVR aggression task, which is only explained at the end of the introduction. Please look into the flow of your introduction to make it more easy to read (for example: choose to use only one set of terms to describe the dichotomy between reactive and proactive aggression, and do this in one paragraph instead of spread around the entire introduction). Furthermore, shorten the introduction considerably (for example the second paragraph (GAM / SIP) could be removed entirely without compromising the key message of the introduction, and the same goes for various other parts of the introduction (the AQ/STAXI descriptions and provocation paradigms could be discussed in less detail. Also, the intervention study using VR could be removed from the introduction, since the present study is not an intervention study (therefore not relevant in this section and, thus, less distracting for the reader), but it may be interesting to mention this study in the discussion as another field of application for VR in relation to aggression).

4. In the introduction: please first write the immersive virtual reality (IVR) and consequently in abbreviated form later (it is first mentioned in abbreviated form only in the first section at page 4)

5. Introduction, page 7, concerning the following sentence: These issues of plausibility are addressed through sensorimotor contingency and domain design, creating a pragmatic method of measuring physical acts of aggression. Could the authors explain how this is done?

6. In the final section of the introduction, be consequent in using past or present tense in describing the present study. And please do not jump to another previous study here (this closing part of the introduction should only describe the present study)

7. Introduction, last paragraph: is the aim of the study actually to develop an assessment? Should this not be: to determine if the (already) developed IVR-assessment is a good (valid) instrument to assess aggression?

8. Method: is it possible for the authors to give more information about the participants? Were these all students? Is it known whether they had problems with aggression in the past? And, if not, is this a study limitation? Should the task not better be assessed in clinical forensic groups who are known for high levels of aggression and where it is most likely to be used in the future? (this latter point is already appropriately discussed by the authors in the discussion section, however)

9. Method, STAXI: please be consequent in present or past tense. Can the anger control scales be explained briefly like the anger expression scales?

10. Method, the IVR task: can the authors give information of the degree of emotional expression on the avatar’s face and posture (in other words: hoe realistic is for example an aggressive or a friendly approach with respect to nonverbal features, apart from the lip synchronisation?)

11. Method: the authors describe that the participant sees a virtual arm to enact the reaction. However, the participants actually use a joystick to initiate this virtual response, which means that their physical sensorimotor perception is not congruent with the perceived visual response. Can the authors elaborate on what in their opinion this might mean for the ecological validity of the task (in the discussion section)?

12. Method, IVR: how is the response action operationalized? Please explain. For the results section: in the correlation and regression analysis it is not clear at all what type of variable is used here in the analyses to represent the IVR-conditions! This must be explained (in the discussion is mentioned that this concerns response time, but this needs to be clear in the previous sections as well). And it needs to be explained why the authors think that this measure is representative of actual aggression. Furthermore, nowhere in this article is described what the direction of the correlation is; only that it is significant. Thus, are shorter reaction times related to higher levels of aggression or are longer reaction times related to higher levels of aggression? And how do the authors explain this finding? (please elaborate on this in the discussion section) What is the relevance of it and is this known from literature?

13. Method, IVR task: can the authors explain what the function is of the congruent versus non-congruent conditions?

14. Procedure: did the authors investigate if the order of the presented blocks in the IVR task was of influence on the outcomes of the aggression questionnaires (in other words: was there a priming effect of the IVR- conditions on the questionnaires – for example when the last condition elicited an aggressive state preceding the questionnaires - and is there corrected for such an effect?)

15. Results, Table 2: What correlations are reported here? Pearson’s r? The RPAQ variables in the first column do not correspond with the RPAQ variables in the top row. (and please add RPAQ to the text above this table where correlations with reactive past aggression are described)

16. Results: last paragraph: please add that Anti-Social 1 was the congruent condition

17. Discussion: can the authors give an indication of the expected field of application for this tool? Do they see it as an instrument for empirical studies only or as an actual diagnostic tool in clinical practice? For example to assess the type and severity of a patient’s aggression problem, or to determine an individual’s improvement during aggression management treatment?

18. Discussion: A strength of the VR environment can be that it is fun to do, which might make individuals more willing to cooperate with such assessments, which is often a problem in forensic populations. The authors might stress this more.

19. Discussion: Might it be a valuable addition for the future to combine the IVR task with biomarkers such as heart rate or skin conductance to even better assess aggressive responses?

20. In the abstract is suggested that IVR is ethically safe. This point is not mentioned in the manuscript. Could it be explained in the discussion section how the authors see this?

21. References: 31, Klain Tuente is spelled incorrect (this should be Klein Tuente)

Reviewer #3: This is an interesting study showing the potential usefulness of IVR for assessment purposes and (potentially) for treatment of aggressive people. The main problem has to do with the sample composition: the authors should move the section 'Participants' to the start of the Results section. Moreover they should give many additional details about the participants: education; occupation; any history of mental disorders or psychological treatment; any history of offenses, etc. Once they have added these variables (hoping that they have this info!!), they should make additional analyses to see whether any of these important sociodemographic features is associated to different IVR responses. Incidentally, the sample is clearly composed by university students, as shown by the young mean age. Moreover, the large majority of the participants are women (30 males out of 116 participants): the authors must comment on this very important sample feature, since a century of research on aggression has shown that aggressive behaviour, of any kind, is largely more frequent and severe in males compared to females. They should have considered this during recruitment and should have tried at least to achieve a fair gender balance in the sample composition. Therefore the over-representation of young females represents a major limitation of this study and has to be acknowledged.

The potential utility of IVR for treatment purposes, including its utilization in clinical populations, has to be demonstrated and so far there is nothing which can support this claim: this has to be made clear.

There are a few typos which have to be corrected:

- page 3: para 2, add 'how' after 'describes'.

- page 5: line 10, drop 'this'.

- page 6: correct 'aggreagable' in 'agreeable'.

- same page, end page: add ' to 'participants'.

- page 7: line 2, 'change'

6. PLOS authors have the option to publish the peer review history of their article (what does this mean?). If published, this will include your full peer review and any attached files.

Reviewer #1: No

Reviewer #2: **Yes: **Niki Kuin

Reviewer #3: No

---

## [Author Response · Author response to Decision Letter 0]

31 Jan 2022

Please see the attached responses to reviewer letter.

---

## [Decision Letter · Decision Letter 1]

16 Feb 2022

PONE-D-21-27929R1Assessing reactive physical violence using Immersive Virtual RealityPLOS ONE

Dear Dr. Terbeck,

Thank you for submitting your manuscript to PLOS ONE. After careful consideration, we feel that it has merit but does not fully meet PLOS ONE’s publication criteria as it currently stands. Therefore, we invite you to submit a revised version of the manuscript that addresses the points raised during the review process.

While the manuscript has been considerably improved, I agree with the Reviewers that further modifications are needed. I encourage Authors to take into account Reviewers' comments for further revision. 

We look forward to receiving your revised manuscript.

Kind regards,

Stefano Triberti, Ph.D.

Academic Editor

PLOS ONE

Journal Requirements:

Reviewers' comments:

Reviewer's Responses to Questions

**Comments to the Author**

1. If the authors have adequately addressed your comments raised in a previous round of review and you feel that this manuscript is now acceptable for publication, you may indicate that here to bypass the “Comments to the Author” section, enter your conflict of interest statement in the “Confidential to Editor” section, and submit your "Accept" recommendation.

Reviewer #2: (No Response)

Reviewer #3: (No Response)

2. Is the manuscript technically sound, and do the data support the conclusions?

Reviewer #2: Yes

Reviewer #3: Partly

3. Has the statistical analysis been performed appropriately and rigorously? 

Reviewer #2: Yes

Reviewer #3: Yes

4. Have the authors made all data underlying the findings in their manuscript fully available?

Reviewer #2: Yes

Reviewer #3: Yes

5. Is the manuscript presented in an intelligible fashion and written in standard English?

Reviewer #2: Yes

Reviewer #3: Yes

6. Review Comments to the Author

Reviewer #2: In general, I think the manuscript is greatly improved compared to the first submission. The authors have addressed all of my previous considerations appropriately. I still have some minor comments, which I explain below. If these points can be adjusted by the authors, I believe the manuscript is suitable for publication in Plos One.

Abstract:

- In the sentence: "In congruent trials, ..."please enter 'the' before 'participant' (or change into 'participants', plural).

- Last sentence: suggest should be suggests

Introduction:

The introduction has definitely been improved, compared to the first submission. It's easier to read and has a better flow to it. There are some minor points to consider:

- p3 the first lines don't fit well together. I suggest the authors remove the sentence 'The General aggression model... lead to aggression.", since there is no further mention of the model anywhere in the paper, and place the following section directly after the definition of aggression (which needs to be placed between '...' by the way, and a '.' needs to be removed at the end of that line).

- p3 the last sentence in this paragraph needs to be clarified with respect to how the reader should interpret 'discrete social dynamics' and 'therefore' (I don't see how the second part of this sentence can logically be derived from the first).

- p4 in the section on the STAXI, the sentence "However, studies have shown mixed results", please add the subject of the studies here

- what is 'impression management'? Please explain

- p5, last paragraph, first line. VR has not yet been implemented as an intervention tool in clinical practice, but only in studies. Please clarify this.

- p6: authors suggest that their task assesses levels of physical reactive aggression. I wonder if you can regard this as a measure of physical aggression if it occurs in a virtual situation (since the virtual environment implies that it is not physical at all!). Would it not be on the safer side to characterize this as 'reactive aggression'?

-p 7. Authors now state much clearer that they use reaction time as a proxy for aggression. However, it is still not clear to me why they assume that a faster reaction time is indicative of aggressive tendencies?

Method:

- p 13, table 1. Why is an agrressive response to a friendly approach (pro-social 2, incongruent) regarded as a measure of inhibition? This is not an inhibitory response in my opinion... Or do authors think that it is inhibitory due to the incongruent nature of the response?

- p 14. last sentence, shorter reaction times are regarded as representing greater levels of aggression. Is this only in the Anti-social 1 condition? Please add this information here

- p15. Participants were told that they could leave at any time when uncomfortable. How many did actually leave?

Results:

- p16: almost half of the participants made errors during the IVR task. What type of errors should we think of here? What does this imply for validity of the test results with respect to reaction time? This point is not yet addressed in the discussion.

- p 16: how do the authors explain that more mistakes were made in congruent than incongruent trials, while you would expect the opposite? Doe this finding need further elaboration in the discussion?

Discussion:

- p 18, first paragraph of the discussion. Please add the direction of the relationship (thus: higher levels of self-reported aggression were correlated to lower reaction times on the IVR paradigm)

- p 20, second paragraph, "Also, since IVR is usually regarded...". please check the grammar of this sentence (since and thus combined seems redundant)

- p 20, bottom section, "This could be achieved by...", before during, after treatment -- > before, during and after treatment

- p 20, last sentence 'be made and to that': remove 'and' here?

References

- p 23, remove [19] between 14 and 15

Reviewer #3: In my previous review of the manuscript I wrote: "THE LARGE MAJORITY OF THE PARTICIPANTS ARE WOMEN (30 MALES OUT OF 116 PARTICIPANTS): THE AUTHORS MUST COMMENT ON THIS VERY IMPORTANT SAMPLE FEATURE, SINCE A CENTURY OF RESEARCH ON AGGRESSION HAS SHOWN THAT AGGRESSIVE BEHAVIOUR, OF ANY KIND, IS LARGELY MORE FREQUENT AND SEVERE IN MALES COMPARED TO FEMALES. THEY SHOULD HAVE CONSIDERED THIS DURING RECRUITMENT AND SHOULD HAVE TRIED AT LEAST TO ACHIEVE A FAIR GENDER BALANCE IN THE SAMPLE COMPOSITION. THEREFORE THE OVER-REPRESENTATION OF YOUNG FEMALES REPRESENTS A MAJOR LIMITATION OF THIS STUDY AND HAS TO BE ACKNOWLEDGED". I was expecting to see this issue clearly mentioned in the section on 'Study limitations' (please add a sub-heading for Limitations). In their rebuttal letter the authors write: "We do agree strongly with the reviewer’s key point that most participants were students and female. This initial sample was used as first validation of the task, but off course future studies would need to apply and replicate the results using forensic sample. We have now provided much more acknowledge of this in the discussion of the paper".

Since a selection bias in the sample composition is always a major threat to study validity, this issue should be clearly discussed and clarified in the Limitation section. Unfortunately, this has NOT been done, and in my opinion the paper should NOT be accepted until I see a clear, detailed para dealing with this major study limitation (selection bias in sample composition).

7. PLOS authors have the option to publish the peer review history of their article (what does this mean?). If published, this will include your full peer review and any attached files.

Reviewer #2: No

Reviewer #3: No

---

## [Author Response · Author response to Decision Letter 1]

12 Apr 2022

Please see the attached "responses_to_reviewers" letter

---

## [Editor Report · Decision Letter 2]

25 Apr 2022

Assessing reactive violence using Immersive Virtual Reality

PONE-D-21-27929R2

Dear Dr. Terbeck,

We’re pleased to inform you that your manuscript has been judged scientifically suitable for publication and will be formally accepted for publication once it meets all outstanding technical requirements.

Kind regards,

Stefano Triberti, Ph.D.

Academic Editor

PLOS ONE
---

## [Editor Report · Acceptance letter]

27 Apr 2022

PONE-D-21-27929R2 

Assessing reactive violence using Immersive Virtual Reality 

Dear Dr. Terbeck:

I'm pleased to inform you that your manuscript has been deemed suitable for publication in PLOS ONE. Congratulations! Your manuscript is now with our production department. 

Kind regards, 

on behalf of

Dr. Stefano Triberti 

Academic Editor

PLOS ONE